# The Thermal Niche of the Koala (*Phascolarctos cinereus*): Spatial Dynamics of Home Range and Microclimate

**DOI:** 10.3390/ani15152198

**Published:** 2025-07-25

**Authors:** Dalene Adam, Carla L. Archibald, Benjamin J. Barth, Sean I. FitzGibbon, Alistair Melzer, Amber K. Gillett, Stephen D. Johnston, Lyn Beard, William A. Ellis

**Affiliations:** 1Wildlife Science Unit, School of Agriculture & Food Science, The University of Queensland, Gatton, QLD 4343, Australia; s.johnston1@uq.edu.au; 2Centre for Integrative Ecology, School of Life & Environmental Sciences, Deakin University, Melbourne Burwood Campus, Burwood, VIC 3125, Australia; c.archibald@deakin.edu.au; 3Koala Ecology Group, School of Agriculture & Food Sciences, The University of Queensland, St. Lucia, QLD 4067, Australia; b.barth@uq.edu.au (B.J.B.); s.fitzgibbon@uq.edu.au (S.I.F.); a.gillett@uq.edu.au (A.K.G.); w.ellis@uq.edu.au (W.A.E.); 4Koala Research Centre, Central Queensland University, Rockhampton, QLD 4701, Australia; a.melzer@cqu.edu.au; 5School of Biological Sciences, The University of Queensland, St. Lucia, QLD 4067, Australia

**Keywords:** climate change, habitat selection, shelter trees, climate refugia, behaviour, ecology, conservation

## Abstract

The koala, an iconic Australian species, is already under threat across much of its distribution and is also particularly susceptible to climate change. Our aim was to understand whether koalas seek out cooler parts of their habitat to cope with high environmental temperatures. Temperature data loggers were set out throughout a section of the koalas’ habitat and their movements were tracked using GPS collars. We found that on hot days, some areas of the koalas’ habitat were up to 10 °C warmer than other areas. During the day, koalas were found in areas of the landscape that recorded lower daytime temperatures, and during the night, they were found in areas that recorded higher daytime temperatures. We suspect that koalas avoided the hot areas during the day by sheltering in non-fodder trees because they provided significant shade and relief from the heat, whereas koalas used these hotter areas during the night because they corresponded with the location of fodder trees. These findings are valuable because they can guide conservation efforts, ensuring both fodder and shelter trees are protected to support koalas’ survival in a warming climate.

## 1. Introduction

Australia is a continent rich in biodiversity, possessing between 7 and 10% of all species on earth with more than half of these being endemic [1,2]. Australia also has more endangered species than any other continent on Earth [2,3]. Many of Australia’s rich and diverse ecosystems are highly sensitive to climate change and it is likely that they will not adapt to the predicted climatic trends [4,5]. There are four main threats to ecosystems under climate change: the influx of new species (native or exotic), altered fire regimes, land-use changes and altered climatic patterns [6]. It is likely that both native and exotic ecological generalist species will expand their range and invade “new” ecosystems, at the expense of native specialists [7]. The most significant impacts of climate change may be the loss of unique fauna and flora due to gradual alterations of ecosystems [6].

The koala is a specialised folivore of *Eucalyptus* [8] and it is recognised by the International Union for Conservation of Nature (IUCN) as a species vulnerable to climate change [4,9,10]. Climate change is expected to act synergistically with existing threats to koalas, particularly heat, drought and fire, causing additional stress on populations across their range [9], leading to increased adverse impacts from disease. However, while some impacts of climate changes are obvious there are other, perhaps more complex, less overt impacts that could equally adversely affect the persistence of koalas. For example, the enrichment of environmental CO_2_ levels accompanying climate change may reduce the availability of nutrients in eucalypt foliage on which the koala depends [9,11,12]. A change in leaf quality will therefore also likely have an effect on koala distribution and survival.

Under realistic future climate change scenarios, Adams-Hosking et al. [4] have predicted that the koala’s distribution will contract significantly eastward and southward. Currently, in Queensland, the koalas in the north and west of their range are probably at the edge of their physiological capacity to endure drought and heat [13]. There have been high recorded mortalities of koalas following prolonged periods of drought coupled with heat waves [9,14]. Several studies [15,16,17] have provided insight into the dynamics of koala populations and their habitat under drought stress on the edges of the koala’s distribution. In doing so, they provide a guide to understanding the nature of drought refugia in central and northern Queensland.

Studies at St Bees Island, found a strong link between periods of extreme temperature and koala deaths [18], which also coincided with low leaf moisture. Koalas consume little free water [19]; as a result they are less able to replenish their water requirements if they need to use evaporative cooling to dissipate heat effectively [13,18,20]. Lunney et al. [9] have indicated that it is vital to understand how trees are used by koalas for feed, water and shelter, so that management and conservation strategies can be put into place to prevent these high mortalities during periods of prolonged drought and heat waves.

Recent observations by Crowther et al. [21] and Briscoe [22] have demonstrated that koalas can make individual tree choices based on nutrition, as well as other factors such as thermoregulation, reiterating the important of climate refugia. Ellis et al. [15] and Melzer et al. [23] provided the theoretical framework to identify refugia. However, despite these observations, there are currently no population data to inform our understanding of the spatial behaviour and whether koalas prefer or select cooler areas, as opposed to individual trees, and relocate in search of cooler areas in times of extreme heat.

Variability in temperature across koala home ranges has not previously been investigated, but it may be an important parameter affecting koala behaviour as much as feed quality, browse moisture and social factors. Our study investigates the spatial dynamics of koalas in the context of temperature and microclimate on St Bees Island. We instrumented the environment and studied koala range utilisation, to detect any spatial correlates with temperature that might inform our understanding of the ability of this species to identify and use climate refugia. The aim of this study was to describe the temporal and spatial variability of temperature experienced by koalas within their range, and assess whether koalas exploit this variability as a means of coping with high environmental temperature. Answering this question will be a key element in understanding whether koalas possess the capacity to exploit environmental variability, and whether they will actively colonise climate refugia as the landscape changes.

## 2. Materials and Methods

### 2.1. Field Site

Early in the 20th century, around the 1930s, koalas were introduced to St Bees Island. St Bees Island is located approximately 30 km off the coast of Mackay, Queensland (20.91899° S, 149.4433° E) and is approximately 1100 ha in size [24]. Today the island remains uninhabited by humans apart from a caretaker. The vegetation has been influenced by historical clearing for grazing and the ongoing browsing of feral goats (*Capra hircus*) and introduced wallabies, swamp wallaby (*Wallabia bicolour*) and whiptail wallaby (*Macropus parryi*). Areas of the island are also covered by grasslands, shrub lands, eucalypt forests, rainforests and low microphyll vine thickets, coastal casuarina woodlands and mangroves. The area where this study was conducted is known as “The Knoll” and is a parcel of land that measures ~20 ha and where koala densities are highest in a low forest dominated by *E. tereticornis* [15]. The island has a humid climate, summers are hot and wet, whereas winters are warm and dry [16,25].

### 2.2. Koalas

Forty-nine koalas were located in 2013 and 2014 by a team of 5–7 experienced koala researchers that systematically searched The Knoll for five consecutive days on field trips in May 2013, August 2013, April–May 2014 and September 2014. Most koalas that were encountered were captured and uniquely ear tagged. Twenty healthy koalas were also collared with GPS units (see details below) and data were collected from 28 August 2013 for approximately 2–2.5 months (see Table 1). Koalas were recaptured in 2014 and the collars removed.

In 2015, as in previous years, systematic transect searches of The Knoll were repeated and all koalas encountered were captured, tagged (if clean skin) and collared with GPS units. A total of 33 koalas were located in 2015 and 2016 on field trips in May, August and December of 2015 and again in May 2016. Six healthy koalas were encountered, captured and collared in December 2015, and GPS data for these koalas was recorded thereafter for approximately 2.5 months (see Table 1). All the koalas caught in 2015 we recaptured in 2016 and their collars were removed.

### 2.3. Koala Capture and Collaring

Koalas were caught using variations on standard capture techniques [26] which typically included climbing the tree and encouraging koalas to descend by flagging or capturing and placing the koala in a cloth bag while in the tree. Koalas were fitted with a custom-made VHF radio collar (two stage transmitter 151 MHz, Titley Scientific Pty Ltd.; Brendale, QLD, Australia). The collars weighed ~120 g and contained a rubber weak link to avoid potential entrapment on tree snags [26]. The collar also included a GPS data logger (iGotU, Mobile Action Technology Inc.; Zhonghe Dist., New Taipei City, Taiwan) positioned at the top of the collar to maximise satellite reception, as previously described by Ellis et al. [26]. To extend battery life and increase the duration of GPS operation, the GPS units were programmed to only record their location every 10 min between 02:00 and 03:00 hrs each day for koalas collared in 2013–2014 (referred to in this manuscript as “night-recorded” GPS locations or home range). For koalas collared in 2015–2016, a GPS location was recorded every 10 min between 13:00 and 14:00 h each day (referred to in this manuscript as “day-recorded” GPS locations or home range). From these data, the single best fix (<15 m accuracy) for each logging period was included for analyses [27,28]. The recording schedule of the GPS loggers was set to start a minimum of one week post-capture to ensure data were not influenced by any abnormal post-capture behaviour.

### 2.4. Microclimate Data

Seventy-six HOBO^®^ pendant temperature and light data loggers (UA-002-64; Bourne, MA, USA) suspended in a modified screen (Hart sport marker, model 44-061, 20.5 cm diameter, 5 cm high) were used to measure ambient temperature (Ta; °C) and relative light levels (W/m^2^). The loggers were deployed across The Knoll (Figure 1), attached to branches of vegetation at a height of approximately 1.5 m above ground in a 50 × 50 m grid pattern (Figure 1). There were no microclimate data recorded during 2013–2014 so that for the koalas whose GPS collars recorded a night GPS location in 2013–2014, the max midday Ta across all the microclimate data from 2015–2016 was used.

### 2.5. Weather Station Data

Solar-powered autonomous HOBO^®^ Micro-station Data loggers (Part # H21-002; Bourne, MA, USA) fitted with a soil moisture probe (m^3^), a weather screen with Ta and relative humidity (RH; %) sensors, and silicon pyranometer (solar radiation, W/m^2^) and tipping-bucket rain gauge were used. There were two weather stations in open grassland (saddle −20.9231; 149.4329) and another in the eucalypt–rainforest woodland (woodland −20.9243; 149.4342). These weather stations recorded climate data continuously. In the current study we utilised climate data from May 2013 to May 2014 and May 2015 to May 2016.

### 2.6. Statistical Analysis

To examine the differences in Ta between weather station data (2013–2014 and 2015–2016) and temperature loggers, a one-way ANOVA (“stats” package, function “aov”; version R 3.6.1, R Core Team 2014; Vienna, Austria) was used. To investigate the specific differences a Tukey test (“stats” package, function “TukeyHSD”; R Core Team 2014; Vienna, Austria) was employed.

The GPS coordinates were uploaded to ArcGIS Pro (version 10.7.1; ESRI Australia) where the decimal degrees (DD) were converted to universal transverse mercator (UTM) coordinates. R software (version R 3.6.1, R Core Team 2014; Vienna, Austria) and the “sp” package was used to convert GPS coordinates (as XY values) into a spatial object in R for further analysis (RStudio version 1.2.5003; R Core Team 2014; Vienna, Austria). The R package “adehabitatHR” was used to estimate kernel home range and the utilisation distribution (UD) of koala home-range size. Setting the kernalUD parameters to the least-square cross validation (LSCV) allowed us to explore home range using the best smoothing value.

The tabulate intersect tool in ArcGIS Pro was used to determine the available microclimate Ta within the koalas’ home range for comparison used microclimate Ta for each koala. Only those koalas whose home range overlapped the study extent by 80% or more were used in the compositional analyses. The R package “adehabitatHS” was used to conduct the compositional analyses using the “compana” function, for each koala [29].

According to Aebischer et al., “an animal’s movements determine a trajectory through space and time; its habitat use is the proportion of the trajectory contained within each habitat type” [29]. If a temperature band is used more than expected from its availability, then it is said to be “preferred”. Compositional analysis was recommended by Aebischer, et al. [29] for the analysis of habitat selection (in the current study = the selection of microclimate) by several animals when there were a number of categories. The analysis was carried out in two steps; (1) the significance of habitat selection was tested (Wilks lambda) and (2) a ranking matrix was built to indicate whether habitat type (in the current study = Ta) in the row is used significantly more or less than the habitat type (the Ta value) in the column. The test parameter in the compositional analysis was set to “randomisation” to determine the differences in log-ratios of utilised habitat composition paired with available habitat composition calculated for each koala home range (available Ta versus used Ta). The habitat type (Ta) was ranked in order of use: highest number which was the most preferred to lowest number which was least preferred. To interpret the analysis; (1) a positive value indicates that Ta in the Ta row is preferred over Ta in the Ta column and visa-versa, whereas (2) a negative value indicates that the Ta in the Ta column is preferred over Ta in the Ta row.

## 3. Results

### 3.1. Environmental Variability: 2015–2016 Microclimate Data

The lowest Ta was recorded in September 2015 (15.2 °C) and the highest Ta was recorded in December 2015 (45.2 °C) (Table 2). The average Ta from August 2015 to May 2016 only varied by 7.6 °C (range: 23.5–31.1 °C). To explore the variation in temperature in the microhabitats, the hottest and coldest days on the island were analysed; the temperature range at midday was used as a reference for comparison. A greater degree of temperature variation occurred across the landscape on the hottest days; at midday (6 January 2016) there was a difference of 10.5 °C between the hottest and coolest available temperatures across The Knoll. The average temperature at midday was 36.6 °C with a range of 31.5–42.0 °C. The coldest day during the study period was 5 September 2015 with a range in temperature at midday of 8.1 °C. On the 11 September 2015, the average daily temperature was 22.0 °C and the max Ta variation across the landscape was as little as 4.0 °C, with the max daily Ta being 24.2 °C.

Habitat types across the study site were classified as open grassy areas with scattered eucalypt species (“open”, average Ta = 36.1 °C) eucalypt forests (“woodland” average Ta = 35.8 °C), rainforest gullies and low microphyll vine thickets (“rainforest” average Ta = 33.9 °C), and coastal casuarina woodlands and mangroves (“shoreline” average Ta = 32.9 °C). Temperature data were grouped by habitat type and averaged; the cooler end of the temperature gradient typically coincided with shoreline habitats and rainforest gullies (Figure 2).

There was a significant difference (*p* < 0.001) in daytime Ta between weather station data collected in 2013–2014, weather station data collected in 2015–2016 and temperature loggers (Figure 3). More specifically, there was a significant difference between logger data (2015–2016) versus 2013–2014 weather stations data (*p* < 0.001) and logger data (2015–2016) versus 2015–2016 weather stations data (*p* < 0.001). There was no significant difference (*p* = 0.27) in Ta between weather station data collected in different years. Averages between weather stations were compared: Ta and humidity were higher in 2015–2016, solar radiation was higher, and the soil probe measured a higher moisture content in 2013–2014.

### 3.2. Koala Home Range

The 95% estimation of kernelUD (LSCV) was used to estimate koala home range. Koala home range varied between 0.6 ha to 16.2 ha for koalas collared in 2013–2014 (2013–2014 n = 20; Table 3). In 2015–2016, koala home range varied between 0.6 ha to 2.6 ha (2015–2016 n = 6; Table 3).

The spatial dynamics of landscape use versus temperature availability varied between koalas. Koalas (n = 10; 2013–2014) whose home range was recorded during the night, appeared to favour the mid to high end of the midday available microclimate temperature range across the landscape (Figure 4). In comparison, koalas (n = 6; 2015–2016) whose home range was recorded during the day appeared to favour the mid to low end of the midday available microclimate temperature range.

### 3.3. Compositional Analysis

Koalas (2013–2014) were more likely to be found during the night in locations that were hotter during the day; 39–43 °C was significantly preferred over 36 °C (Table 4). Although koalas (2015–2016), whose home range was recorded during the day, preferred 36 °C over 31 °C (*p* ≤ 0.05) and 39 °C over 38 °C (*p* ≤ 0.05), koalas appear to prefer mid to low microclimate Ta across the available temperature band (Table 5).

## 4. Discussion

Koalas’ (*Phascolarctos cinereus*) are a threatened species and the ongoing decline in distribution and abundance is compounded by the added pressures of climate change [30,31,32]. The aim of this study was to determine drivers of temporal and spatial variability within the koalas’ home range on the region known as The Knoll on St Bees Island. St Bees Island is remote and accessible only by small boat, so field research is logistically challenging. We did not have the resources to instrument the island in 2013–2014 with Ta data loggers to measure microclimate. In December 2015 we captured far fewer koalas when compared to 2013–2014 and we also did not have access to as many GPS data loggers. However, the study site was chosen because we had access to a historical database to inform our monitoring programme, a detailed understanding of the landscape, the land management regime and resident koala population.

Our results suggest that ambient temperature variation was greatest during the hotter months, suggesting that the benefit of access to cooler microhabitats in summer could be a driver of habitat selection. Furthermore, the differences in daytime Ta recorded by the temperature loggers versus the weather station data highlight the importance of using microclimate data when examining the thermal niche of a species, because the data recorded by a weather station is likely to be significantly different to what the koala is actually experiencing within the various areas of its habitat.

Pfeiffer et al. [33] investigated tree use by koalas on St Bees Island, and by direct observations found that koalas utilised fodder species most frequently at night and non-fodder trees were most frequently utilised during the day. These researchers hypothesised that koalas selected fodder trees at night under a “drive” to feed and selected non-fodder day trees under a “drive” to seek shelter which presumably provided a physiological advantage [33]. Briscoe et al. [20] proved that non-fodder tree trunk temperatures in south-eastern Australia were cooler and that koalas were observed “hugging” the trees as means of conductive heat loss. Similarly, our results demonstrated that the cooler end of the temperature gradient typically coincided with shoreline habitats and rainforest gullies which were mainly made up of non-fodder species.

Kavanagh et al. [34] and Ellis et al. [13] reported that cooler non-fodder trees are likely to be important for buffering koalas against climate extremes. Consequently, it is not surprising that koalas (2015–2016, day-recorded GPS locations) appeared to favour the mid to low end of the available temperature range during the day. In comparison, the night-recorded home ranges of koalas appeared to favour the mid to high end of the available midday temperature range across the landscape. We postulate that the night-recorded home ranges were logged when koalas were active and utilising fodder species, trees that likely do not offer as much respite from the heat when compared to cooler non-fodder trees (*Casuarina* spp. and rainforest species).

Behavioural thermoregulation and habitat selection is not unique to koalas; most animals can modify behaviour to mitigate at least some of the threats posed by warmer temperatures. In North America, moose (*Alces alces*) which are a heat-sensitive ungulate, were found to reduce movement, use areas of the landscape with more shade, and travelled closer to mixed forests and bogs during periods of heat [35]. In the North-Western Italian Alps the Alpine ibex (*Capra ibex*), a wild goat, was found to occupy areas that were less productive, and therefore consumed lower quality forage, when it was warmer; this species prioritised thermoregulation over fodder quality [36]. Similarly, another study conducted on moose found that the moose adjusted their behavioural patterns, in response to heat, by seeking shelter in mature coniferous forest leading to a decrease in local forage availability [37]. These studies demonstrate that habitat structure and microclimate heterogeneity directly influence wild animal behaviour and highlight the importance of thermal shelters in the animals’ ability to persist in warmer temperatures.

In the case of koalas, defining habitat and developing models for future distribution places heavy emphasis on preferred fodder tree species [38,39,40]. However, Callaghan et al. [41] found more faecal pellets under non-eucalypt species than eucalypt, which is likely due to the fact that koalas are utilising non-eucalypt as roosting trees; these researchers acknowledge the importance of “shelter” trees but place heavy emphasis on koala occurrence and the availability of preferred fodder species. More recently the importance of non-fodder trees as a climate refuge has gained traction [13,17,42]. However, the historical oversight on the importance of shelter trees, as well as the lack of incorporating physiological data [43], in mapping prime koala habitat and some of the techniques used (e.g., koala surveys only occurring in eucalypt stands) have influenced models which predict the future distribution of koalas and the impact of climate change on this species. While fodder species are obviously important, we propose that climate refugia are equally important in determining koala habitat and should also be incorporated (alongside physiological and behaviour data) into models used to accurately predict the future distribution of koalas.

## 5. Conclusions

This study is the first to investigate koala microclimate use at a moderate spatial scale by combining landscape-level temperature mapping with GPS-based home-range analysis. Our results showed that there was a large variation in temperature across the landscape and that koalas were able to exploit this variability to cope with the high temperatures. While our data collection was conducted under a number of limitations and assumptions, our analysis, nevertheless, provides insights into the different functions of fodder and non-fodder trees. As conservationists and managers we have to employ flexible conservation strategies [44] and rethink how we classify koala habitat, which then has direct implication to understanding habitat retention and revegetation strategies for maintaining and developing climate change refugia for this species. For example, integrating both fodder and non-fodder trees in restoration projects, preserving mixed-species forests, maintaining wildlife corridors, and ensuring forest health are all critical factors for the long-term persistence of koalas. Furthermore, it highlights the importance for further investigations into the physiological limits of the koala’s thermal niche to more accurately inform management strategies and to predict the impact of climate change on this species.

## Figures and Tables

**Figure 1 animals-15-02198-f001:**
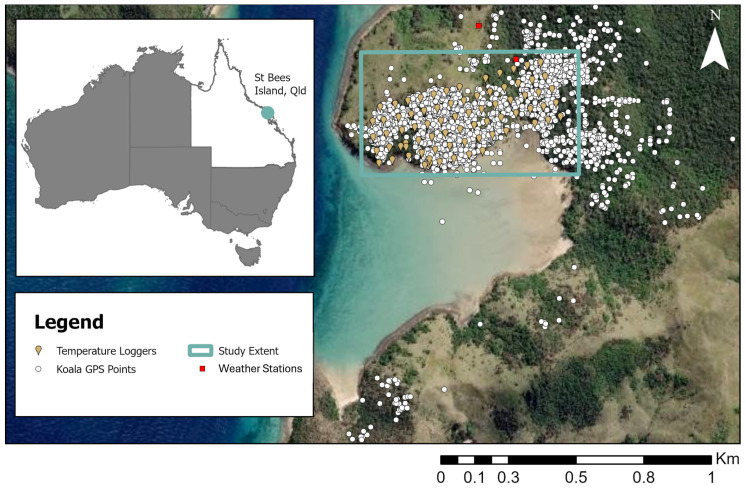
Study site: a map (ArcMaps) showing the layout of the microclimate temperature loggers (*n* = 76) across The Knoll, the distribution of koala GPS points and weather stations on St Bees Island during 2013–2014 and 2015–2016. Insert: map of Australia depicting location of St Bees Island, off the coast of Mackay, Queensland.

**Figure 2 animals-15-02198-f002:**
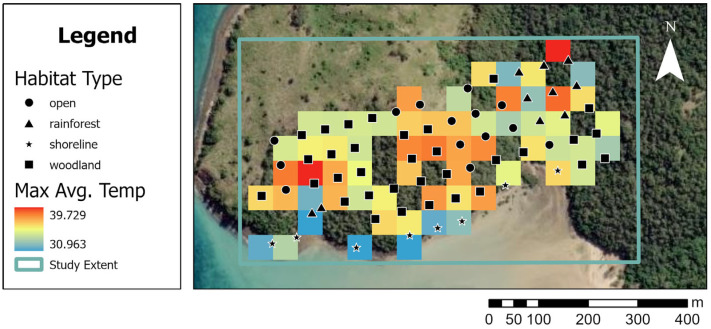
A map (ArcMaps) showing the layout of the microclimate temperature loggers (*n* = 76) across The Knoll (St Bees Island) with habitat type layered over available temperature polygons across the landscape of the study extent (see Figure 1) on the hottest day (6 January 2016) at midday.

**Figure 3 animals-15-02198-f003:**
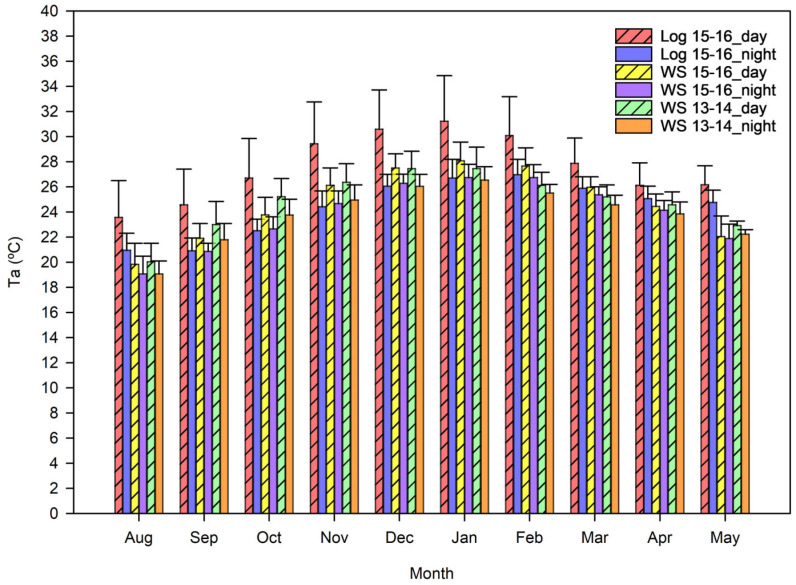
Bar graph depicting the comparison between average weather station (WS) data from 2013–2014 and 2015–2016 and temperature loggers (Log) microclimate data 2015–2016. Error bars represent standard deviation.

**Figure 4 animals-15-02198-f004:**
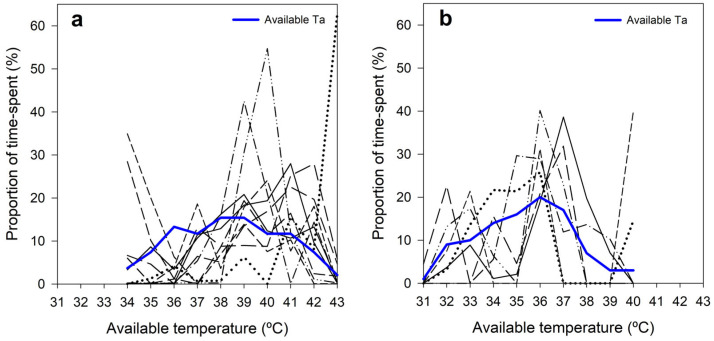
Available microclimate temperature (X axis) plotted against the proportion of time spent within that temperature band across the landscape in the study extent (Y axis) (see Figure 1), (**a**) calculated from koala 2013–2014 night-recorded home-range data and (**b**) 2015–2016 day-recorded home-range data. Microclimate data used in this analysis was the absolute max Ta at midday (2013–2014 koalas) and Ta recorded on the hottest day at midday (2015–2016 koalas).

**Table 1 animals-15-02198-t001:** Koalas captured, collared and monitored in 2013–2014 and 2015–2016: koala ID (St Bees Island koala male or female 1, e.g., SBM1 or SBF1), sex and weight.

Koala ID	Sex	Weight (kg)	Monitored 2013–2014	Monitored 2015–2016
SBM1	Male	8.7	Yes	No
SBM2	Male	5.6	Yes	No
SBM3	Male	6.7	Yes	No
SBM4	Male	8.8	Yes	No
SBM5	Male	7.3	Yes	No
SBM6	Male	8.5	Yes	No
SBM7	Male	4.7	Yes	No
SBM8	Male	8.6	Yes	No
SBM9	Male	7.3	No	Yes
SBM10	Male	5.1	No	Yes
SBF1	Female	4.2	Yes	No
SBF2	Female	6.6	Yes	No
SBF3	Female	8.4	Yes	No
SBF4	Female	6.6	Yes	Yes
SBF5	Female	6.9	Yes	Yes
SBF6	Female	6.5	Yes	No
SBF7	Female	6.4	Yes	No
SBF8	Female	8.0	Yes	No
SBF9	Female	6.5	Yes	No
SBF10	Female	8.0	Yes	No
SBF11	Female	6.3	Yes	No
SBF12	Female	7.9	Yes	No
SBF13	Female	4.3	No	Yes
SBF14	Female	5.3	No	Yes

**Table 2 animals-15-02198-t002:** The absolute min and max and average microclimate temperatures (day–night) across The Knoll recorded by the Ta loggers (*n* = 76). Shaded according to the temperature: the higher the temperature the darks the red is.

	Min (°C)		Max (°C)		Ave (°C)	
	Day	Night	Day	Night	Day	Night
August 2015	16.8	17.1	37.9	25.2	23.5	20.9
September 2015	15.2	15.6	38.7	24.1	24.4	20.8
October 2015	17.1	17.2	40.6	25.0	26.5	22.4
November 2015	20.7	19.9	43.6	27.5	29.2	24.2
December 2015	21.9	22.1	45.2	29.0	30.4	25.9
January 2016	21.2	20.9	42.9	30.9	31.1	26.5
February 2016	22.2	22.1	44.1	31.0	30.0	26.8
March 2016	21.6	21.9	39.2	28.7	27.8	25.9
April 2016	21.0	20.3	37.9	27.9	26.1	25.1
May 2016	22.0	22.1	37.3	27.9	26.2	24.7

**Table 3 animals-15-02198-t003:** Koala ID, night-recorded home-range estimates for koalas collared in 2013–2014 and day-recorded home-range estimates for koalas collared in 2015–2016, and an indication (yes or no) of whether the koala data were included in the compositional analysis (if home range overlapped the study extent by 80% or more).

Koala ID	Home Range (ha) 2013–2014	Home Range (ha) 2015–2016	Comp. Analysis	Year Collared
SBM1	2.1	-	Yes	2013–2014
SBM2	1.0	-	Yes	2013–2014
SBM3	10.2	-	No	2013–2014
SBM4	8.4	-	No	2013–2014
SBM5	6.1	-	Yes	2013–2014
SBM6	1.3	-	No	2013–2014
SBM7	12.7	-	No	2013–2014
SBM8	3.0	-	Yes	2013–2014
SBM9	2.3	2.3	Yes	2015–2016
SBM10	0.6	0.6	Yes	2015–2016
SBF1	16.2	-	No	2013–2014
SBF2	7.1	-	No	2013–2014
SBF3	1.8	-	Yes	2013–2014
SBF4	2.3	1.8	Yes	2013–2014 and 2015–2016
SBF5	1.4	1.53	No: 2013–2014 Yes: 2015–2016	2013–2014 and 2015–2016
SBF6	0.7	-	No	2013–2014
SBF7	4.9	-	Yes	2013–2014
SBF8	1.4	-	Yes	2013–2014
SBF9	2.8	-	Yes	2013–2014
SBF10	0.6	-	No	2013–2014
SBF11	3.4	-	Yes	2013–2014
SBF12	1.9	-	No	2013–2014
SBF13	-	2.6	Yes	2015–2016
SBF14	-	0.9	Yes	2015–2016

**Table 4 animals-15-02198-t004:** Differences in log-ratios calculated from koala 2013–2014 night-recorded home-range data comparing midday-recorded microclimate temperature preferences (based on landscape use by koalas) with availability (based on temperature logger study extent, see Figure 1). Asterisks (*) indicates significant difference (*p* ≤ 0.05) in log-ratios between used and available Ta. The “Rank” column indicates most to least preferred temperature bands (increments of 1 °C), with the most preferred temperature band being scored a 9 to lease preferred being scored a 0.

Ta (°C)	Differences In Log-Ratios Matrix										Rank
	34	35	36	37	38	39	40	41	42	43	
34	0	-	-	-	-	-	-	-	-	-	3
35	−0.85	0	-	-	-	-	-	-	-	-	1
36	−1.67 *	−0.82	0	-	-	-	-	-	-	-	0
37	−0.77	0.08	0.9	0	-	-	-	-	-	-	2
38	0.04	0.89	1.71 *	0.81	0	-	-	-	-	-	4
39	1.30	2.14 *	2.96 *	2.06 *	1.25 *	0	-	-	-	-	9
40	0.98	1.83 *	2.65 *	1.75	0.94	−0.31	0	-	-	-	8
41	0.92	1.77 *	2.59 *	1.69	0.88	−0.37	−0.06	0	-	-	7
42	0.92	1.76	2.58 *	1.69 *	0.87	−0.38	−0.07	−0.01	0	-	6
43	0.27	1.12	1.93 *	1.04	0.22	−1.02	−0.72	−0.66	−0.65	0	5

**Table 5 animals-15-02198-t005:** Differences in log-ratios calculated from koala 2015–2016 day-recorded home-range data comparing midday-recorded microclimate temperature preferences (based on landscape use by koalas) with availability (based on temperature logger study extent, see Figure 1). Asterisks (*) indicates significant difference (*p* ≤ 0.05) in log-ratios between used and available Ta. The “Rank” column indicates most to least preferred temperature bands (increments of 1 °C), with the most preferred temperature band being scored a 9 to lease preferred being scored a 0.

Ta (°C)	Differences in Log-Ratios Matrix										Rank
	31	32	33	34	35	36	37	38	39	40	
31	0										3
32	0.80	0									7
33	0.40	−0.39	0								6
34	0.24	−0.56	−0.16	0							5
35	−0.39	−1.19	−0.80	−0.63	0						1
36	1.98 *	1.19	1.58	1.74	2.38	0					9
37	1.05	0.25	0.64	0.81	1.44	−0.93	0				8
38	−0.85	−1.65	−1.26	−1.09	−0.46	−2.84	−1.90	0			0
39	−0.24	−1.03	−0.64	−0.48	0.16	−2.22	−1.28	0.62 *	0		2
40	0.12	−0.68	−0.28	−0.12	0.51	−1.86	−0.93	0.97	0.36	0	4

## Data Availability

The corresponding author will supply the relevant data in response to reasonable requests.

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
