# Peer review of "The Thermal Niche of the Koala (Phascolarctos cinereus): Spatial Dynamics of Home Range and Microclimate"

_animals, 2025, doi:10.3390/ani15152198_

Round 1

Reviewer 1 Report

Comments and Suggestions for Authors

Manuscript ID: animals-3662503

Title: The thermal niche of the koala (Phascolarctos cinereus): spatial dynamics of home range and microclimate

by Dalene Adam et al.

Review

The objective of the present study was to describe the temporal and spatial variability of temperature within koala home ranges and to assess whether koalas exploit this variability to cope with high environmental temperatures. Study was designed to ascertain whether koalas possess the capacity to utilize microclimate variation and actively seek out climate refugia in response to environmental change.

As shown by Briscoe, N. J. (2015). Tree-hugging behavior beats the heat. Temperature, 2(1), 33-35, koalas use tree trunks, particularly non-feeding Acacia mearnsii, as thermal refugia during heat. Trunk surfaces were ~5 °C cooler than ambient air, and koalas responded by moving lower and pressing their bellies to these cooler spots, reducing the need for evaporative cooling. The study highlights fine-scale temperature variation within trees and links koala posture to thermal profiles, though it does not assess movement or home-range data.

In reviewed manuscript, Adam et al. introduces a more comprehensive methodological approach compared to N. Briscoe’s earlier work by integrating GPS-based movement data with a high-resolution microclimate grid to analyze koala behavior across the landscape (Briscoe, 2015). While Briscoe demonstrated that koalas use cool tree trunks, her study was limited to individual tree-scale observations using thermal imaging and surface temperature measurements, without tracking broader movement patterns. In contrast, Adam and colleagues deployed 76 temperature loggers across St Bees Island in a structured grid and combined this with daytime and nighttime GPS tracking of koalas to assess their spatial preferences in relation to ambient temperature. Their analysis included habitat classification, statistical modeling of temperature preferences, and compositional analysis of habitat selection, revealing that koalas actively seek cooler microclimates during the day, especially in non-fodder trees, and shift to warmer areas at night for feeding. This approach allowed for a more detailed understanding of how koalas exploit thermal variability at the home-range scale, highlighting the critical role of shelter trees in climate adaptation.

Compared to Crowther, M. S., Rus, A. I., Mella, V. S., Krockenberger, M. B., Lindsay, J., Moore, B. D., & McArthur, C. (2022). Patch quality and habitat fragmentation shape the foraging patterns of a specialist folivore. Behavioral Ecology, 33(5), 1007-1017, authors of reviewed manuscript measures, not inferrers thermal refuge via tree size/canopy. Adams et al. present direct microclimate or operative temperature data, they integrate spatial use and inferred thermal shelter, as well as microclimate selection.

Therefore, while both shown references are worth to cite, reviewed manuscript is better in methodological approach and statistical data treatment. I have some points to comment for revision, to increase readability, but I have no critical issues to report.

General comments

  1. Use x–y notation with en-dash to represent a span or range of numbers, dates, or time, e.g., 7–10 % in Line 51, [9,12–14] in Line 69, 2013–2014 in Line 175, and the rest of text. The same for page ranges in References.
  2. Subchapters in Material and Methods have wrong numbering, it must start from 2
  3. Figure 1 should go to chapter 2.1.
  4. Part of text from 3.3. should go to Methods, to explain thermal preference.
  5. First paragraph of discussion should be condensed, not to repeat data already shown, it can summarize obtained results.
  6. Discussion could be wider, including not only koalas. Field studies demonstrate that fine-scale habitat structure and in situ microclimate heterogeneity directly influence wild mammal behavior, habitat selection, and thermoregulation, with the strongest evidence coming from research on American pikas, large ungulates and elephants, not only koalas, using paired behavioral observations and microclimate sensors.
  7. Back matter, namely conflict of interests require to indicate,, if funders had any influence on … (check Template).
  8. Used format “doi: https://doi.org/10.1071/WR10156” is wrong. Using both "doi:" and the "https://" prefix together is redundant and non-standard. The format should be consistent with the citation style guidelines.

Specific comments

  1. Select keywords less similar to the Title.
  2. Lines 113–114, accuracy is ca. 1m? not too detail?
  3. Table 1: body score was not explained in Methods.
  4. Line 181, caption of Figure 1 could start as “study site”. And the inset explained in the end.
  5. Figure 3: utilization of analogous green and yellow shades in the bar chart presents a discernibility issue, particularly for individuals with color vision deficiencies. The overlap in hue between categories such as WS 15–16_day, WS 15–16_night, and WS 13–14_day reduces visual clarity, while the hatching patterns intended to differentiate between night and day readings are not distinct enough in scale or direction. Furthermore, the predominant dark green employed in Log 15–16_day captures attention in a disproportionate manner when juxtaposed with the more muted tones. These issues compromise the readability and accessibility of the data, making it difficult to interpret temperature trends across time and data sources. A more efficacious approach would entail the utilization of distinctly contrasting colors from disparate families (e.g., blue, orange, and purple) in conjunction with simplified, consistent patterns. This method would facilitate the differentiation of variables, thereby ensuring clarity.
  6. Tables 3 and 4 could be merged, adding new column Period or Year
  7. Figure 4 is hardly readable, and I think accent is wrong – Ta should be more visible, while individual differences do not need animal number. So, both panels can fit side to side across page width. (a) and (b) can be used to mark both panels, explanation from the top of each panel moved to caption. Also, please clear mistype in Legend.
  8. First sentence of Conclusions could be “This study is the first to investigate koala microclimate use at a moderate spatial scale by combining landscape-level temperature mapping with GPS-based home-range analysis”.

Author Response

Thank you very much for reviewing this manuscript. 

Please find attached a copy of my response. 

Kind regards, 

Dalene 

Reviewer 2 Report

Comments and Suggestions for Authors

Abstract:  During the day koalas were found in areas of the landscape that recorded lower daytime temperatures and during the night were found in areas that recorded the highest daytime temperatures. – Line 39-41 spatial use during the night is related to day-time temperature?

The abstract is well structured and well written.

The structure of the simple summary does not follow the abstract and is written quite complicated. If you follow the structure of the abstract and formulate it more simply it would be fine

Introduction

Line 55-58 You are talking about climate change as threat at first and then describe the four main threats. I would it turn around.

Line 66-67 I agree that stress is related to increased adverse impacts from disease but is this the main problem? I assume that the habitat changes too and this means a decrease of the population

Line 68-74 I understand the thought but the line of argumentation is not clear. Looking at tree species is not enough, there are different functions and there are different qualities even within the same species.

Line 75-85 This is the same for this paragraph. You are talking first about the special situation at St. Bees followed by more general assumptions which relate to the paragraph above. Connect the two paragraph and point on the special situation in St. Bees Island afterwards.

The following paragraph would perfectly fit and I suggest to move line 93-97 to the discussion.

I am fine with the last paragraph.

Material and method

Field site – connect the first sentence to the sentence in line 121. Be more specific – what means early in the 20thcentury? Wet and hot summer, warm and dry winter? Specify with numbers

3.3. Koala capture and collaring

Is it correct that for 2013/14 there are only night time fixes and for 2015/16 only day time fixes?

Line 172-178 should be shifted to the discussion

Fig. 1: In my opinion there is no need to show the Koala fixes. I would relate the study outline to the raster in Fig. 2 and show only the position of the logger. Show the positions of the weather stations in the figure too.

I am fine with the description of the statistical analysis. In my opinion there is no need to describe how you converted the coordinates and the use of best fix was already described in line 157. Describe it only one time.

3.1. Environmental variability: 2015-2016 microclimate data

I would describe first the results in Tab. 2 and then pinpoint on the variation in temperature during the warmest and hottest day. I cannot see the data you describe in line 236 in Tab. 2.

Tab. 2 what do the colours mean?

Line 247 - the cooler end of the temperature gradient typically coincided with shoreline habitats and rainforest gullies – is this supported by statistics. Looking at Fig. 2 I cannot see it.

Fig. 2 Do you show the average for the whole recording period or for the hottest day only (as described). If yes why the hottest day only and what average?

In the legend two digits after the decimal point should be fine.

Fig. 4 I suggest to use the same extent of the X-Axis for both graphs

Discussion

Line 320-333 describes “only” the results and should be shifted to the result section.

Line 334- Investigations into Ta of habitat type showed that the cooler end of the temperature 334 gradient coincided with shoreline habitats and rainforest gullies which were mainly made 335 up of non-fodder species. I miss an analysis to support this statement

In the discussion you focus on Koala only but what you describe is behavioural thermoregulation which appears in many species. I would like to see a discussion in this context. You describe, that Koala favour the mid to high end of the available midday temperature range during the night. Why is daytime temperature related to habitat use during the night?

The weakest point of the study is the combination of temperature and location data of different years. I miss a discussion of this aspect. Maybe you can prove be comparing the weather station data that temperatures between the years do not differ significantly.

I am not sure whether you need to stress climate change in the discussion. If you do, describe the possible consequences.

Conclusion

While our data collection was conducted under a number of limitations and assumptions, our analysis, nevertheless, provides valuable comparisons between night-recorded home-range and day-recorded home-ranges – this has to be proven

The conclusion is very general and could focus on the main result – different functions of fodder and non-fodder trees and the consequences for Koala conservation.

Author Response

Thank you very much for reviewing this manuscript. 

Please find attached the responses. 

Kind regards, 

Dalene 

Round 2

Reviewer 1 Report

Comments and Suggestions for Authors

Thank you for revised version, but still, there are some minor comments on it.

Comment: Use x–y notation with en-dash to represent a span or range of numbers, dates, or time, was not fully acknowledged: Lines 103, 163, 176, 178, 193, 194, 195, 209, 265, 292, 294, 298, 303, 307, 308, 310, 311, across Table 3, 319, 321, 328, 329, 330, 340, 399, and all page ranges in References.

Similarly, en-dash is required in Figure 2.

Comment: coordinates of the island are (–20.91899°S, 149.4433°E), revise Lines 137, 216, 217